# Longitudinal Phenotypes Improve Genotype Association for Hyperketonemia in Dairy Cattle

**DOI:** 10.3390/ani9121059

**Published:** 2019-12-01

**Authors:** Francisco A. Leal Yepes, Daryl V. Nydam, Sabine Mann, Luciano Caixeta, Jessica A. A. McArt, Thomas R. Overton, Joseph J Wakshlag, Heather J. Huson

**Affiliations:** 1Department of Population Medicine and Diagnostic Sciences, College of Veterinary Medicine, Cornell University, Ithaca, NY 14853, USA; fal43@cornell.edu (F.A.L.Y.); dvn2@cornell.edu (D.V.N.); sm682@cornell.edu (S.M.); jmcart@cornell.edu (J.A.A.M.); 2Department of Veterinary Population Medicine, College of Veterinary Medicine, University of Minnesota, 1365 Gortner Ave, Saint Paul, MN 55108, USA; lcaixeta@umn.edu; 3Department of Animal Science College of Agricultural and Life Sciences Cornell University, Ithaca, NY 14853, USA; tro2@cornell.edu; 4Department of Clinical Sciences, College of Veterinary Medicine Cornell University, Ithaca, NY 14853, USA; jw37@cornell.edu

**Keywords:** hyperketonemia, dairy cow, GWAS, longitudinal phenotype, BHB, NEFA

## Abstract

**Simple Summary:**

Dairy cows have differing success in supporting their physiological functions while in energy deficit right after calving. Identification of genomic regions associated with different concentrations of non–esterified fatty acids and β–hydroxybutyrate in early postpartum Holstein cows provide insight into an animal’s genetic susceptibility to these conditions. Longitudinal phenotypes may provide a different perspective than cross-sectional phenotype variation and their association with genotypes in the study of complex metabolic diseases in dairy cows. This might allow us to reinforce preventative measures that decrease the incidence of hyperketonemia and improve genetic selection criteria.

**Abstract:**

The objective of our study was to identify genomic regions associated with varying concentrations of non-esterified fatty acid (NEFA), β-hydroxybutyrate (BHB), and the development of hyperketonemia (HYK) in longitudinally sampled Holstein dairy cows. Our study population consisted of 147 multiparous cows intensively characterized by serial NEFA and BHB concentrations. To identify individuals with contrasting combinations in longitudinal BHB and NEFA concentrations, phenotypes were established using incremental area under the curve (AUC) and categorized as follows: Group (1) high NEFA and high BHB, group (2) low NEFA and high BHB), group (3) low NEFA and low BHB, and group (4) high NEFA and low BHB. Cows were genotyped on the Illumina Bovine High-density (777 K) beadchip. Genome-wide association studies using mixed linear models with the least-related animals were performed to establish a genetic association with HYK, BHB-AUC, NEFA-AUC, and the comparisons of the 4 AUC phenotypic groups using Golden Helix software. Nine single-nucleotide polymorphisms were associated with high longitudinal concentrations of BHB and further investigated. Five candidate genes related to energy metabolism and homeostasis were identified. These results provide biological insight and help identify susceptible animals thus improving genetic selection criteria thereby decreasing the incidence of HYK.

## 1. Introduction

The ability of a cow to deal with extensive physiological changes during the late pregnancy and early postpartum period influences the entire lactation in terms of milk yield, quality, and health status [1]. Strong genetic selection of dairy cows, driven by the ability to achieve high milk production, particularly in early lactation, has increased the early postpartum gap between energy consumed and energy required [2]; however, some cows can overcome this crucial phase of metabolic adjustments when the energy demand is doubled immediately after calving [3] without negative sequela for the animal’s production or health. 

During this period of negative energy balance (NEB), cows respond by mobilizing lipid and protein from tissue reserves in order to compensate for the reduced intake of nutrients [1] and these reserves are used to support lactation and vital functions [4]. Fat reserves are released into the blood stream as non-esterified fatty acids (NEFA) that can be extracted and metabolized by several body tissues such as skeletal muscle, liver, and kidney. Some cows adapt very well to NEB; however, other cows do not, resulting in excessive ketone body synthesis. Dairy cattle produce three different ketone bodies: acetoacetate, acetone, and β--hydroxybutyrate (BHB). The BHB concentration in blood has been widely used to diagnose hyperketonemia (HYK) in dairy cattle [5]. A low correlation between NEFA and BHB concentrations during the transition period was reported previously in cross-sectional [6] as well as longitudinal studies [7,8]. This shows that some cows seem to effectively use NEFA in the adaptation to lactation while having low BHB concentrations, whereas other cows exhibit excessive ketone body synthesis.

Hyperketonemia is considered one of the most complex diseases in dairy cattle because there are many factors involved in its development such as advanced parity, increased body condition score before calving [9], nutrition during the dry period [10], over-crowded pens [11], and environment [12]. Up to 40% of cows on a typical dairy farm will be hyperketonemic, and the peak of HYK incidence occurs at 5 days in milk (DIM) [13]. Hyperketonemia has been associated with increased metabolic diseases and reduced milk production [14,15,16]. The cost per individual cow case of HYK in the U.S. has been estimated to be $289 accounting for all the direct and indirect costs associated with the disorder [17]. Heritability of HYK or concentrations of energy metabolites in blood in dairy cows has been reported to range from 0.02 to 0.39 [18,19,20,21]. The difference between the heritability reported in previous studies may be due to differences in the characterization of HYK and degree of NEB during early lactation in dairy cows. 

A genome-wide association study (GWAS) allows us to analyze in detail the relationship between genotypic and phenotypic data, thereby associating single-nucleotide polymorphism (SNP) allelic frequencies to disease [22]. A genetic analysis can identify the heritability of quantitative traits that are risk factors for the disease [23]. However, the use of longitudinal as an alternative to cross–sectional phenotypes in genetic studies may provide a better understanding of the genetic mechanisms influencing the onset and progression of complex metabolic diseases. We hypothesized that longitudinal phenotypes could improve the ability to detect genetic association with complex metabolic diseases such as HYK in dairy cows. Our intensively characterized dataset allowed us to isolate genetic differences of cows within similar environmental and management conditions. Therefore, the objective of our study was to identify genomic regions associated with the development of HYK (BHB ≥ 1.2 mmol/L) based on serial or single concentrations of NEFA and BHB over time in early postpartum Holstein cows.

## 2. Materials and Methods 

### 2.1. Longitudinal Phenotype Collection

All procedures were approved by the Cornell University Institutional Animal Care and Use Committee (protocols nos. 2008–0099 and 2011–0016). Our study population consisted of 147 Holstein dairy cows from two trials. Both trials evaluated multiparous cows having serial measurements of serum/plasma NEFA and blood BHB concentration from calving until 16 DIM 3 times per wk. Blood samples were collected from coccygeal vessels. Serum was tested for NEFA concentration (HR Series NEFA–HR (2), Wako Life Sciences, Mountain View, CA) and BHB using a cow side test (Precision Xtra meter, Abbott Diabetes Care Inc, Alameda, CA, USA). This cohort of cows was not treated for hyperketonemia during either of the trials. Detailed information about the first study evaluating 63 cows was previously reported [24]. An extra blood sample (7 mL) was harvested once using evacuated glass tubes (Beckton Dickinson Vacutainer System, Franklin Lakes, NJ) with K_3_ EDTA and a 20 gauge × 2.54 cm blood collection needle for subsequent DNA extraction. After collection, blood tubes were gently inverted 5 times to homogenize blood with K_3_ EDTA and immediately placed on ice to prevent DNA degradation. After that, blood samples were then stored at −20 °C until DNA extraction and subsequent genotyping. The second trial similarly evaluated 84 cows [25]. DNA was extracted from muscle biopsies which were performed for all cows within the study [26]. The muscle biopsies were placed in liquid nitrogen and stored at −80 °C until DNA extraction and subsequent genotyping. 

### 2.2. DNA Extraction and Genotyping

Whole blood or muscle tissues were submitted for DNA extraction and genotyping to GeneSeek laboratories (Lincoln, NE, USA). The DNA extraction was performed with Omega Mag Bind Tissue kit for DNA extraction following the manufacturer’s instructions (OMEGA bio-tek, Norcross, GA, USA). Whole-genome genotypes of 777,962 SNPs were generated using the Illumina Bovine High-density beadchip [27]. Genotype data were filtered using Golden Helix SNP & Variation Suite (SVS) 8.3.4 software (Golden Helix, Bozeman, MT, USA). Genetic quality control was performed excluding SNPs with a call rate <0.90, minor allele frequency (MAF) <0.05 or if the number of alleles was ≥2. A total of 521,929 SNPs remained for analysis after quality control filtering. Sample quality control was performed, and 19 samples (12.9%) with a call rate <0.90 were excluded. 

### 2.3. Phenotypic Classification 

There is a lack of information regarding the genetic factors predisposing cows to HYK due to the complexity of the disease. The aforementioned trials from which these biological samples derived allowed for a unique opportunity to investigate these genetic parameters using 6 serial measurements of BHB and NEFA concentration. Multiple phenotypic classifications were established to identify variation in the genetic regulation of HYK. First, HYK was defined as concentration in blood of BHB (≥1.2 mmol/L) at a minimum of one time point from calving until 16 DIM, corresponding to previous characterization of HYK [15,28].

A second, broader approach was also taken by calculating the incremental area under the curve (AUC) using the 6 concentrations of serum/plasma NEFA or blood BHB from the longitudinally sampled cows from calving until 16 DIM. The trapezoidal rule was used to estimate AUC by summing the area of all the trapezoids formed between two time points [29] with the statistical software package SAS 9.4 (SAS Institute Inc.; Cary, NC, USA). The AUC allowed us to compile the serial measurements from each individual into a single continuous variable and preserve the variability within the dataset as opposed to using either a single measure of the given trait, e.g., HYK, or cow average for the trait. We expected that phenotypic misclassification in our study population would be lower using this approach.

All modern dairy cows will face NEB during early lactation and some will mobilize body reserves as NEFA to a greater extent, whereas others to a lesser extent. Therefore, ketogenesis magnitude will differ among cows as reflected by circulating BHB. These parameters were used to categorize cows into groups that contrast their ability to mobilize as well as metabolize energy reserves. The resulting AUC were used to group cows to identify individuals with the most variation during the first 16 DIM. For BHB area under the curve (BHB-AUC), a high concentration was defined at values >7.2 mmol/L; this threshold was generated by computing the AUC of 6 single measurements of BHB >1.2 mmol/L [24]. For NEFA area under the curve (NEFA-AUC), a high concentration was considered at values >4.2 µEq/L; this value was set by calculating the AUC of 6 single measurements of NEFA of >0.7 µEq/L [6]. Therefore, in our study, all cows were then classified into 4 different phenotype groups based on their NEFA-AUC and BHB-AUC as follows: Group (1) high NEFA and high BHB (*n* = 10), group (2) low NEFA and high BHB (*n* = 11), group (3) low NEFA and low BHB (*n* = 69), and group (4) high NEFA and low BHB (*n* = 57). The variables BHB-AUC and NEFA-AUC were not normally distributed, therefore their values are given as median and range. The variables BHB-AUC and NEFA-AUC were analyzed using the nonparametric Kruskal-Wallis test with PROC NPAR1WAY (SAS 9.4, SAS Institute, Cary, NC, USA). Chi-square tests using PROC FREQ (SAS 9.4) were performed to identify differences among the variables farm, parity, and HYK when phenotype groups were used as the response variable.

### 2.4. Genome-Wide Association Study

The degree of relatedness between pairs of cows in this study was computed to identify highly related animals using genomic identity-by-descent (IBD) estimations in SVS 8.3.4 software (Golden Helix, Bozeman, MT, USA). The IBD estimates the likelihood of specific alleles being inherited from a common ancestor when comparing two individual samples. IBD estimates allowed for the identification and removal, if necessary, of highly related animals in lieu of pedigree information to minimize the risk of false positives results [22] (Figure 1).

Genome-wide association studies were performed to establish associations between low frequency SNP variants and the development of HYK or different levels of BHB and NEFA in early postpartum Holstein dairy cows. The Efficient Mixed Model Association eXpedited (EMMAX) algorithm is a mixed linear model embedded in Golden Helix software which corrects for population stratification and relatedness [30] and was used to perform the GWAS analysis. This model was the most suitable because the population structure is taken into account by including the kinship matrix previously generated with the IBD procedure. The kinship matrix was included as the variance–covariance structure of the random effect for the individuals [31]. The single-locus mixed model GWAS EMMAX uses the following general equation, y=Xβ+u+e, where *y* is an *n* × 1 vector of observed phenotypes, and *X* is an *n* × *q* matrix of fixed effects including mean, SNPs, and other confounding variables. Βeta is a *q* × 1 vector representing coefficient of the fixed effects. U is the random effect of the mixed model with $\mathrm{Var}\left(u\right)\;=\;\sigma_{g}^{2}K$, where *K* is the kinship matrix inferred from the genotype and e represents the error term and is the residual that cannot be explained by the variables in the model [32]. HYK was evaluated as a categorical variable with individuals designated as case, equating HYK diagnosis, or control. Similarly, a pair-wise evaluation was conducted by comparing designated groups having differential BHB-AUC and NEFA-AUC measures to one another. Thereby, Group 1 was independently compared to each remaining Group in separate GWAS. We note that the GWAS comparing Group 1 to Group 2 was not performed due to the extremely low number of individuals in both Groups (*n* = 11, *n* = 10, respectively). Lastly, the measures of BHB-AUC and NEFA-AUC were analyzed independently as continuous variables in their respective GWAS. Parity, farm, milk production, and disease events (i.e.; displaced abomasum, metritis, and retained placenta) were evaluated for inclusion in the model. Specific diet and dry matter intake were unavailable. Health events were excluded due to inconsistency in farm recording. Milk production was excluded due to an incomplete dataset and subsequent unbalancing of the model. Parity and farm were retained as fixed effects in all GWAS. Multiple testing correction using the false discovery rate (FDR) was performed to diminish the probability of Type I error. The FDR was calculated using the formula $\mathrm{FDR}\sim \frac{1}{k}\overset{K}{\underset{i=1}{\sum }}\mathrm{Pr}(H_{0_{i}}|y)$, where *K* is equal to the number of SNPs used on the final examination [33,34]. Candidate genes were located by referencing 0.5 Mbp up- and 0.5 Mbp down-stream from the significantly associated SNPs passing multiple testing correction using the University of Maryland (UMD) 3.1 bovine genome assembly. Genes which showed biological plausibility in hyperketonemia were highlighted for discussion. 

Pseudo-heritability or narrow-sense heritability was calculated for HYK, BHB-AUC, and NEFA-AUC using the formula $h^{2}\;=\;\sigma_{g}^{2}/\left(\sigma_{g}^{2}+\;\sigma_{e}^{2}\right)$, were h^2^ is the response heritability, $\sigma_{g}^{2}$ is the genetic variance, and $\sigma_{e}^{2}$ is the estimate of environmental variance. The GWAS analysis partitions the observed phenotypic variance into the additive genetic and nongenetic components. This estimation can be used to determine heritability, also known as pseudo-heritability [35,36]. Pseudo-heritability variance was estimated using SVS (Golden Helix) Software, based on the algorithms reported by Yang et al. [37].

### 2.5. Haplotype Analysis

Linkage disequilibrium (LD) was calculated using the expected maximization (EM) logarithm to derive *r*^2^ estimates of pair-wise LD using the SVS software. Linkage disequilibrium was computed to identify haplotype blocks and their potential association with the phenotypic variables. Haplotype analysis enhances the information obtained from the association test by incorporating the information from multiple markers among the same gene or genes with minimal historic recombination. The expectation maximization (EM) algorithm embedded in Golden Helix SVS software was used to compute haplotype frequencies. The EM algorithm is an iterative technique that starts with arbitrary values (expectation step), and these values are used to calculate the haplotype frequencies by maximum likelihood (maximization step). This algorithm allows us to perform automatic detection of LD block through the whole genome. 

Haplotype Trend Regression (HTR) takes one or more blocks of genotypic markers and for each block of markers, estimates haplotypes and then regresses their by-sample haplotype probabilities against a dependent variable. Haplotype analysis is an important part of association testing as it can be sensitive to unmeasured variants which may be missed in a single SNP analysis [38]. It can also provide an alternative marker panel consisting of a series of consecutive markers, therefore less sensitive to the effects of recombination on prediction accuracy for use in genomic selection.

## 3. Results

Twenty-nine percent of the individuals (*n* = 42) were defined as HYK at a minimum of one time point from calving until 16 DIM in our dataset. We confirmed that the misclassification of HYK in our study population is lower using the serial measures as opposed to one opportunity afforded by cross-sectional studies. This is evident in that the percentage of individuals defined as HYK at a single time point ranged from 7% (time point 1) to 15% (time point 3). Indeed, 36% of the HYK individuals (*n* = 15 out of 42) only had a single measure of BHB concentration ≥1.2 mmol/L out of the six time points and were, therefore, probable candidates for misclassification if this were a cross-sectional study. In contrast, 26% of the HYK individuals (*n* = 11 out of 42) were likely to be diagnosed as HYK regardless of time point, given that they had four or more elevated measures of BHB (≥1.2 mmol/L). By using the average BHB concentrations for individuals, only 12% of the study cohort (*n* = 17) would have been diagnosed HYK. The variation in HYK designation when comparing single time points as given in a cross-sectional study or average BHB concentrations to our definition of HYK (minimum of 1 elevated BHB) showcases the relevance of using serial measures for phenotypic classification of hyperketonemia.

To further refine our phenotypic characterization of hyperketonemia, we calculated area under the curve for the serial BHB and NEFA concentrations. This allowed us to distinguish individuals who had a single elevated concentration of BHB or NEFA, respectively, from those who had two, three, four, five, or six episodes of elevated BHB or NEFA. It also incorporated concentration variation into the calculated AUC variable, thereby distinguishing individuals demonstrating particularly high concentrations (i.e.; BHB 3.9 mmol/L) from those with lower concentrations including the HYK-defined minimum threshold of 1.2 mmol/L. The NEFA-AUC ranged from 0.40 to 9.03 µEq/L over 16 d with a median of 3.92 µEq/L, and the BHB-AUC ranged from 1.60 to 14.25 mmol/L over 16 d with a median of 3.65 mmol/L (Figure 2). The distribution of parity, HYK, and farm was also analyzed, and total counts are shown in Table 1 based on the group designation.

The Q-Q plots from the different mixed linear model using EMMAX are shown in Appendix A. These plots showed most of the observed −log_10_ (*p*-Value) following a uniform distribution, indicating that our genetic quality control was appropriate. Moreover, Q-Q plots are showing few uncorrected log_10_ transformed *p*-values located in the tail of the plots with a significant deviation of the expected uncorrected log_10_ transformed *p*-values, in agreement with the results obtained from the different Manhattan plots.

We performed 8 different GWAS using HYK as a dichotomous phenotype, BHB-AUC as a continuous phenotype, NEFA-AUC as a continuous phenotype, and the 5 possible pair-wise combinations of the 4 AUC phenotype groups. Figure 3 shows the Manhattan plots of HYK, BHB-AUC, and NEFA-AUC using the uncorrected log_10_ transformed *p*-values. Results for HYK and BHB-AUC are very similar, which is to be expected with HYK diagnosis being dependent upon BHB concentrations. That being said, BHB-AUC results show consistently higher degrees of association for the majority of the significant SNPs, which is likely reflective of using the area under the curve approach encompassing a greater degree of the variation from the 6 serial measures. This can be seen with markers on chromosomes 4, 5, 8, 10, 16, and X. The NEFA-AUC GWAS did not provide any significant associations after multiple testing correction. The GWAS reflecting the pair-wise comparisons of the categorical groups is shown in Figure 4. Similar to HYK and BHB-AUC results, the categorical grouping gave the most promising results when comparing high BHB concentration groups (Groups 1 & 2) to low BHB concentration groups (Groups 3 & 4) (Figure 4a,b,d,e). No significant associations were identified when only comparing variation in NEFA concentrations (Figure 4c). Despite the similar general outcomes reflecting BHB concentrations, this categorical approach shows genomic variation potentially related to NEFA mobilization as seen by markers on chromosome 3 in Figure 4a as opposed to Figure 4b. These GWAS compare Groups 1 or 2, both having high BHB-AUC partnered with either high NEFA-AUC or low NEFA-AUC, respectively, to Group 3, which reflects animals with both low BHB and NEFA-AUC. 

Nine SNPs (Table 2) passing multiple testing correction (FDR ≤ 0.05) were explored for candidate genes with a biological relationship with changes in NEFA, BHB, or the development of HYK during the early lactation period. All genes found within 1 Mb of the associated SNP corresponding to the bovine reference sequence or human genome annotation are identified in Table 2. Five genes were identified within this group as the most plausible candidate genes affecting HYK based on their functional annotation. Hydroxysteroid (17-beta) dehydrogenase 10 (*HSD17B10*), ATP-binding cassette transporter 1 (*ABCA1*), and hepatic lipase (*LIPC*) genes were identified on chromosomes X, 8, and 10, respectively, using the bovine reference genome. The 5-hydroxytryptamine (serotonin) receptor 2C, G protein-coupled (*HTR2C*), and ATP-binding cassette transporter 2 (*ABCA2*) genes were identified on chromosomes X and 8, respectively, using gene homology to human and mouse assemblies. 

In our study, the pseudo-heritability of HYK was 0.16 ± 0.56, of BHB-AUC was 0.82 ± 0.44, and of NEFA-AUC was 0.02 ± 0.14. Standard errors for pseudo-heritability are likely magnified due to the small sample size yet reflect similar estimates of previous studies. Haplotype analysis showed no significant association after testing for haplotypes frequencies with HTR (*p*-value = 0.65). In all, significant associations were identified on four chromosomes and highlighted five biologically plausible candidate genes. The small sample size likely limited potential findings, while the use of the serial NEFA and BHB measures mitigated this issue by improving the accuracy of the phenotypic characterization. 

## 4. Discussion

The objective of our study was to assess the ability of longitudinal phenotypes to improve detection of genetic association with complex metabolic diseases such as HYK in dairy cows. A multitude of GWA studies have focused on production, phenotype, and health traits, but only a small portion of these specifically investigate transition cows or tackle complex metabolic disorders. While our study had a relatively small sample size, reducing the power to find all true genomic associations, a similarly powered study of 73 individuals identified a QTL for Holstein cholesterol deficiency [39] and an in-depth report on small sample size GWAS in dogs showed the effectiveness of just 20 dogs for mapping traits within breed [40]. More importantly, the improved phenotypic characterization through serial measurements of NEFA and BHB concentration in blood during the first 16 DIM provides an advantage by providing a higher phenotypic reliability [21]. Indeed, simulation studies have found that a mere 10% misclassification of phenotype reduces the reliability of correctly identifying predictive SNPs in a GWAS to 54% [41,42]. In addition, the use of a high-density SNP panel having markers spanning the genome is more desirable for identifying novel genomic regions associated with complex traits or diseases.

Frequently, there are multiple SNPs associated with complex diseases such as HYK, and each one can increase the risk of developing the diseases in small increments. As the understanding of HYK, energy metabolism, and the transition cow period is improved, we are afforded the opportunity to advance our knowledge of their genetic regulation. To date, one gene-based study combined with pathway analysis identified various biological pathways associated with NEFA, BHBA, and glucose changes in cows sampled 3 weeks before expected calving, 4 weeks postpartum, and 13 weeks after parturition [43]. Preliminary data by Kroezen et al. [44] identified a panel of 1081 SNPs associated with HYK based on producer-recorded cases of clinical HYK to be tested in a larger cohort of Canadian cattle. The same group found different regions associated with mid-infrared spectroscopy-predicted milk BHB concentrations in Holstein dairy cows [45]. Zoetis, a private company offering genotyping and genomic prediction services for dairy cattle, released their Wellness Traits, including ketosis predictions, in 2016 [46]. The Council on Dairy Cattle Breeding is now offering a similar genomic prediction for breeding merit of ketosis susceptibility as part of the U.S. National Dairy Cattle Genomic Evaluations as of April 2018 [47]. 

Hyperketonemia is a complex disorder that has many potential risk factors, including genetic factors [33]. Accuracy of the diagnosis of HYK and the intricacies of BHB and NEFA concentration variation play a crucial role in identifying genomic regions associated with this disease. The in-depth phenotyping with the longitudinal sampling during the first 16 DIM allowed for a more accurate assessment of HYK as opposed to single measurement derived from a cross-sectional study design, and mitigated the effect of the small sample size. Here, we present 5 candidate genes with biological relevance in the development of HYK or with high AUC for BHB and NEFA concentrations that were identified in multiple GWAS comparing 521,929 SNPs from 128 least-related Holstein cows with longitudinal measures.

### 4.1. Candidate Genes

Hydroxysteroid (17-beta) dehydrogenase 10 (*HSD17B10*). This gene is located on chromosome X from 96,267,144–96,269,467 base pairs (bp) (RefSeq: NM_174334.3) and is well conserved in all vertebrates. In 4 out of 8 GWAS, *HSD17B10* emerged as a candidate gene: Hyperketonemia (dichotomous), BHB-AUC (continuous), group 2 vs. group 3 (dichotomous), and group 2 vs. group 4 (dichotomous). Yang et al. [48] reported that *HSD17B10* gene encodes for a mitochondrial multifunctional enzyme, which catalyzes the oxidation of steroid modulators of gamma aminobutyric acid type A receptors and steroid hormones. It also has short-chain 3-hydroxy-2-methylacyl-CoA dehydrogenase activity, an essential step in the degradation of isoleucine. Isoleucine is an essential branched chain amino acid (EAA) that plays a pivotal role in protein and energy metabolism [49] with particular importance due to the contribution to milk protein synthesis [50]. Mutations in *HSD17B10* have been reported in humans and caused a complete loss of a mitochondrial multifunctional enzyme which were biochemically diagnosed with an elevated concentration of metabolites from isoleucine breakdown [51]. Given *HSD17B10*’s role in energy metabolism and protein synthesis, we hypothesize that this gene’s activity and efficiency may play a role in hyperketonemia.

The 5-hydroxytryptamine (serotonin) receptor 2C, G protein-coupled (*HTR2C*). *HTR2C* gene is located on chromosome X from 67,986,710–68,083,180 bp (RefSeq: AC_000187.1). In 6 out of the 8 performed GWAS, *HTR2C* was identified as a candidate gene: HYK (dichotomous), BHB-AUC (continuous), group 1 vs. group 3 (dichotomous), group 1 vs. group 4 (dichotomous), group 2 vs. group 3 (dichotomous), and group 2 vs. group 4 (dichotomous). The hypothalamus consolidates all the processes related with energy homeostasis. The ability of transition cows to gain energy homeostasis is vital as they manage NEB and move towards neutral and positive energy balance which relates to the relevant events and key time period of HYK. The α-melanocyte stimulating hormone (α-MSH) is produced by pro–opiomelanocortin (POMC) neurons in the arcuate nucleus (ARC) of the hypothalamus. The α-MSH is an agonist of melanocortin 4 receptors (MC4Rs) and the melanocortin signaling mediates food intake, body weight, and energy metabolism [52]. The effect of *HTR2C* might be crucial for POMC neuronal activation [53]. In addition, the activation of the central serotonin system has been linked with depressed appetite in almost all mammals [54]. Rodent models with a deletion of *HTR2C* showed hyperphagia and obesity [53]. In humans, manipulation of 5-*HTR2C* receptors in the hypothalamus has been used to effectively induce weight loss using drugs such as d–fenfluramine and phentermine via blocking the reuptake of serotine and prompt its release [55,56,57]. Moreover, HTR2C effect on POMC might be influenced by circulating energy metabolites such as glucose, fatty acids, leptin, and insulin [56,58]. 

ATP-binding cassette transporter *ABCA1* and *ABCA2*. *ABCA1* is located on chromosome 8 from 96,274,035–96,390,357 bp (RefSeq: NM_001024693.1). *ABCA2* (RefSeq: AC_000168.1), annotated in the human, mouse, and rat genomes, shows homology to this same region as well. In 5 out of the 8 performed GWAS, *ABCA1* and *ABCA2* appeared as candidate genes: Hyperketonemia (dichotomous), BHB-AUC (continuous), group 1 vs. group 3 (dichotomous), group 1 vs. group 4 (dichotomous), and group 2 vs. group 4 (dichotomous). The adenosine triphosphate (ATP)-binding cassette (ABC) membrane transporter gene superfamily binds and hydrolyzes ATP to move different nutrients (amino acids, lipids, lipopolysaccharides, etc.) across the extracellular and intracellular membranes such as the endoplasmic reticulum (ER) [59,60], peroxisome, and mitochondria [61,62,63]. Mutations within the *ABCA1* gene in humans cause total or partial reduction of normal high-density lipoprotein (HDL) cholesterol due to the role of ABCA1 in HDL formation and in reverse cholesterol transport [61,62,63,64,65]. Subjects with this genetic variation accumulate cholesterol droplets in their liver, spleen, lymph nodes, intestine, and nervous system [60,64]. In the same way, ABCA2 over expression has been correlated with decreased efflux and diminished esterification of lipoproteins [66]. Functional annotation and the association of *ABCA1* and *ABCA2* to health disorders suggests that these genes may play a role in the movement of nutrients needed for energy metabolism and the possible build-up of lipids in the liver related to hyperketonemia.

Hepatic Lipase (*LIPC*). This gene is located on chromosome 10 from 51,758,867 bp to 51,921,040 bp (RefSeq: NM_001035410.1). In 4 out of the 8 performed GWAS, *LIPC* as a candidate gene: HYK (dichotomous), BHB-AUC (continuous), group 2 vs. group 3 (dichotomous), and group 2 vs. group 4 (dichotomous). The *LIPC* gene encodes for a lipolytic enzyme synthesized in the liver that plays a pivotal role in several steps of lipoprotein metabolism [67,68]. Up-and-down regulations have been associated with dyslipidemia; however, the pathways are not well understood [69]. Cohen et al. [70] estimated that the genetic variations at the *LIPC* gene might explain up to 25% of the total variation of HDL plasma concentrations in human monozygotic twins. Moreover, *LIPC* variants might have a pleiotropic effect on one more of the abnormalities associated with metabolic syndrome in humans such as insulin resistance [71]. *LIPC*’s role in lipoprotein metabolism in the liver suggests it may be particularly relevant in the re-esterification of NEFA in the liver. 

### 4.2. Heritability

The pseudo-heritability of HYK was 0.16 ± 0.56 in our study and is congruent with a previous report of 0.17 [19] and greater than the heritability reported by others at 0.02 [18], and 0.06 [72]. Pseudo-heritability is defined as the fraction of phenotypic variance explained by the relationship matrix IBD. However, pseudo-heritability for some traits may over- or under-estimate heritability due to missing heritability, the proportion of genetic variance that cannot be explained by all significant SNPs [35,36]. The difference among these values could be attributed to the higher accuracy of disease characterization and its definition in the study of van der Drift et al. [19] and our own. For our study, the pseudo-heritability of BHB-AUC (0.82 ± 0.44) is a more rigorous estimate than the pseudo-heritability of HYK. The calculation used for pseudo-heritability of BHB-AUC included all serial measurements, therefore the magnitude of standard error is smaller than that of HYK at one time point. While the standard error and standard deviation of pseudo-heritability are high given our small sample size, the pseudo-heritability estimates provide a baseline to compare characteristics within the study [36]. Some of the standard confidence intervals are over the normal limits (0,1) due to the relatively modest sample size. Therefore, we recommend caution with the interpretation of the pseudo-heritability result obtained in our study. 

## 5. Conclusions

Our study differed from previous attempts to identify genomic regions associated with the development of HYK by using serial measurement data from dairy cows during the high risk period of HYK, thus reducing the risk of phenotypic misclassification. Our positive results, despite a small cow cohort, demonstrates the informativeness of serial measures of BHB for the genomic analysis of hyperketonemia as supported by another recent study showing increased genomic prediction accuracy of HYK when using serial measures. In all, the 5 novel candidate genes of *HSD17B10*, *HTR2C*, *ABCA1*, *ABCA2*, and *LIPC* were identified based on genome-wide association to either HYK status or serial blood concentrations of BHB and NEFA. Further confirmation of these regions and candidate genes using an unrelated population and/or expression studies is needed to establish the complete effect of them on HYK in early postpartum Holstein dairy cows. The suggestive results for NEFA concentrations also warrant additional studies in a larger cohort of animals which may provide insight towards the genetic regulation of fat mobilization for energy metabolism. Overall, this study provides a foundation to explore the genetic regulation of HYK and proposes markers for consideration in genomic selection schemes.

To conclude, HYK is one of the most important postpartum metabolic diseases in dairy cattle because of the negative association with reproduction, milk production, and metabolic and infectious diseases and, thus, profitability and health of dairy cows. Our ability to identify the most susceptible animals has been constrained by the ability to measure metabolites such as BHB and NEFA during the high risk period early postpartum when HYK is detected. Genetic studies have been similarly limited by the complexity of the phenotypic characterization of HYK. With these results and future validation in a larger population, the early identification of animals most susceptible to developing HYK using genomic information will provide producers with the ability to selectively breed for healthier animals and intensify the prophylactic measures for those deemed at risk.

## Figures and Tables

**Figure 1 animals-09-01059-f001:**
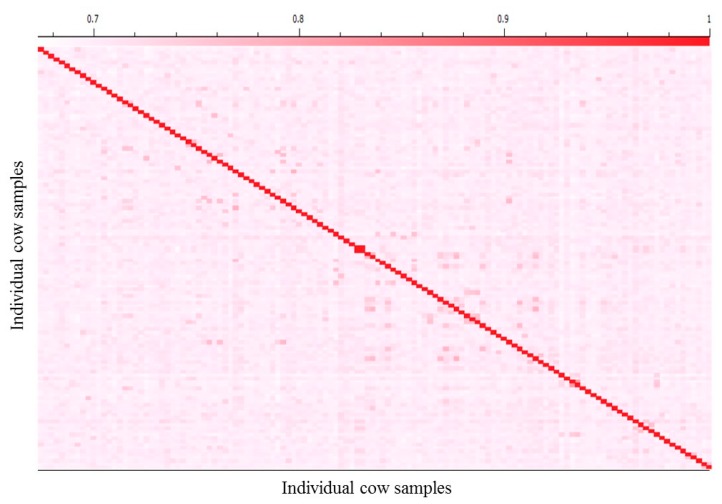
A heat map depiction of the genomic kinship matrix showing the relatedness between each cow sample and general population structure. This matrix is calculated using the identity-by-descent (IBD) procedure from SVS (Golden Helix, Bozeman, MT, USA) Software. The x- and y-axes correspond to the 128 least-related cow individuals. Highly related cows were removed to minimize the risk of false positive results.

**Figure 2 animals-09-01059-f002:**
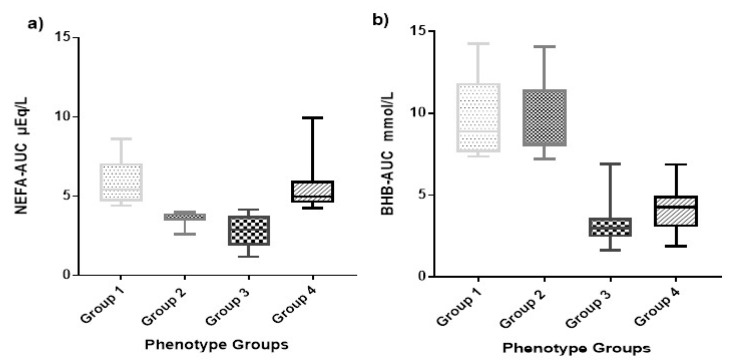
Non-esterified fatty acids (NEFA) area under the curve (AUC) (NEFA-AUC) and β-hydroxybutyrate (BHB) area under the curve (BHB-AUC) box and whisker plot. (**a**) The box and whisker plot showing the distribution of NEFA-AUC µEq/L during the first 16 days in milk (DIM) by phenotype group. (**b**) The box and whisker plot showing the distribution of BHB-AUC mmol/L during the first 16 DIM by phenotype group. The phenotypes groups are: (1) High NEFA-AUC and high BHB-AUC, (2) low NEFA-AUC and high BHB-AUC, (3) low NEFA-AUC and low BHB-AUC, and (4) high NEFA-AUC and low BHB-AUC.

**Figure 3 animals-09-01059-f003:**
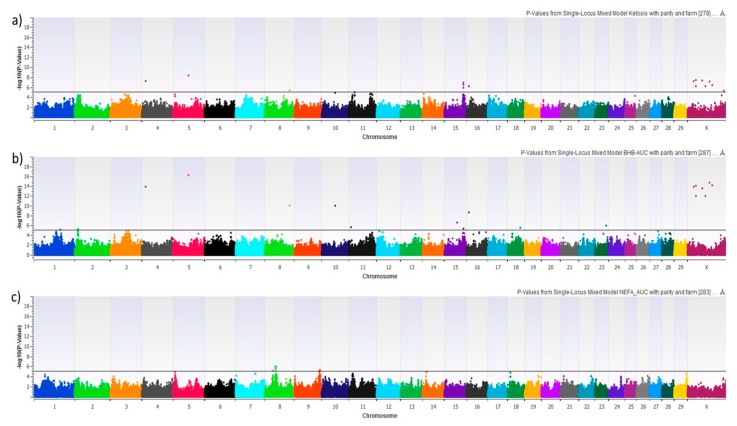
Manhattan plot showing the graphic representation of −log 10 (*p*-Value) from mixed linear models with parity and farm as fixed effects: (**a**) Hyperketonemia, (**b**) BHB-AUC, and (**c**) NEFA-AUC. SNPs above the horizontal black lines achieved a false-discovery rate corrected *p*-value of <0.05 and were explored for biological significance.

**Figure 4 animals-09-01059-f004:**
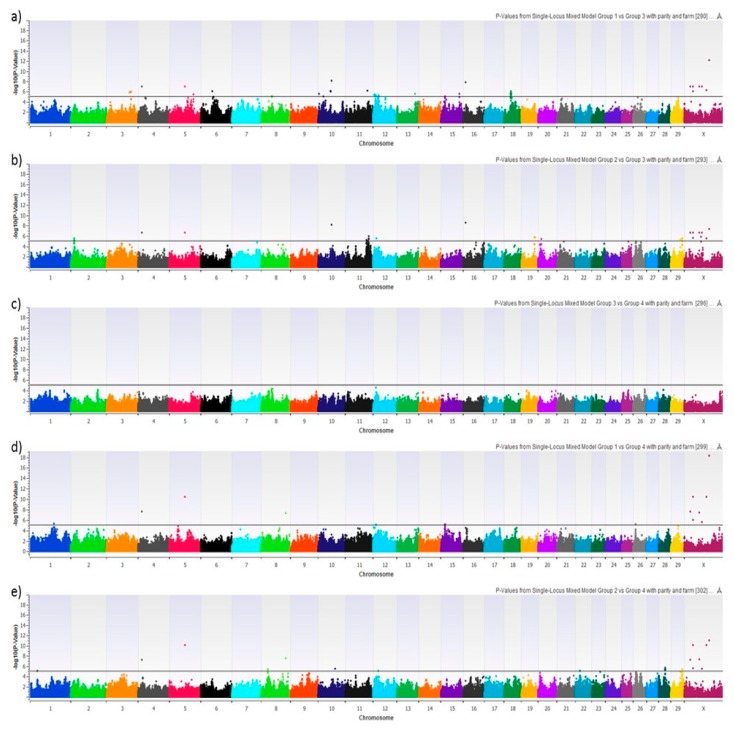
Manhattan plot showing the graphic representation of −log 10 (*p*-Value) from each different mixed linear model. Specifically, this plot corresponds to the mixed linear model with parity and farm as fixed effects: (**a**) Phenotype group 1 (high NEFA-AUC and high BHB-AUC) vs. phenotype group 3 (low NEFA-AUC and low BHB-AUC), (**b**) phenotype group 2 (low NEFA-AUC and high BHB-AUC) vs. phenotype group 3 (low NEFA-AUC and low BHB-AUC), (**c**) phenotype group 3 (low NEFA-AUC and low BHB-AUC) vs. phenotype group 4 (high NEFA-AUC and low BHB-AUC), (**d**) phenotype group 1 (high NEFA-AUC and high BHB-AUC) vs. phenotype group 4 (high NEFA-AUC and low BHB-AUC), and (**e**) phenotype group 2 (low NEFA-AUC and high BHB-AUC) vs. phenotype group 4 (high NEFA-AUC and low BHB-AUC). SNPs above the horizontal black lines achieved a false-discovery rate corrected *p*-value of <0.05 and were explored for biological significance.

**Table 1 animals-09-01059-t001:** Descriptive statistics of the study population. Results are presented as total counts or range of values and median value.

Groups ^1^	Parity/Farm	1	2	3	4	Overall Count	*p*-Value
Total per group		10	11	69	57	147	
Parity number ^2^	2	1	4	18	12	35	0.003
	3	1	3	37	25	66	
	≥4	8	4	14	20	46	
Hyperketonemic ^3^		10	11	5	16	42	<0.0001
BHB–AUC ^4^		8.85 (7.3–14.2)	8.1 (7.2–14)	2.95 (1.6–6.9)	4.25 (1.9–6.9)	3.65 (1.6–14.2)	<0.0001
NEFA–AUC ^5^		5.41 (4.4–8.6)	3.84 (2.6–4)	2.89 (1.16–4.1)	5 (4.2–9.9)	4 (1.16–9.9)	<0.00001
Farm ^6^	1	4	4	11	11	30	0.05
	2	0	4	12	17	33	
	3	6	3	46	29	84	

^1^ Group (1) high non-esterified fatty acids (NEFA) and high BHB, group (2) low NEFA and high BHB, group (3) low NEFA and low BHB, and group (4) high NEFA and low BHB. ^2^ Difference in parity between the groups was calculated with Chi-square. ^3^ Hyperketonemia was the number of cows in each group with a single measurement of BHB ≥1.2 mmol/L during the first 16 DIM and the difference was calculated using Chi-square. ^4^ BHB-AUC are shown as the range of values and median value. The difference in BHB-AUC was calculated with nonparametric Kruskal-Wallis test. ^5^ BHB-AUC are shown as the range of values and median value. The difference in BHB-AUC was calculated with nonparametric Kruskal-Wallis test. ^6^ Farm is expressed as counts and was analyzed with Chi-square.

**Table 2 animals-09-01059-t002:** Genome-wide association study results identifying associated single-nucleotide polymorphisms (SNPs) and candidate genes from the 128 least-related Holstein cows.

Chr	SNP Location ^1^	Gene Name	Gene Start ^2^	Gene End ^2^	Distance to the SNP (bp) ^3^	Best *p*-Value ^4^	False Discovery Rate *p*-Value	Phenotypes Associated by GWAS ^5^
5	60045785	*KIAA0748*	60276249	60300244	254,459	5.98 × 10^−17^	3.12 × 10^−11^	HYK, BHB-AUC, 1 vs. 3, 1 vs. 4, 2 vs. 3 and 2 vs. 4
		*AMOTL2*	60276249	60300244	254,459			
		*TESPA1*	60265872	60301998	256,213			
		*NEUROD4*	60223354	60234099	177,569			
8	95966003	*ABCA1, ABCA2*	96270791	96408375	304,788	1.14 × 10^−10^	5.96 × 10^−06^	HYK, BHB-AUC, 1 vs. 3, 1 vs. 4 and 2 vs. 4
		*OR13F1*	95761537	95762465	42,726			
		*NIPSNAP3A*	96239004	96252150	273,001			
10	51462618	*LIPC*	51758867	51921040	296,249	1.14 × 10^−10^	5.45 × 10^−06^	HYK, BHB-AUC, 2 vs. 3 and 2 vs. 4
		*MYO1E*	51020004	51240914	221,704			
		*CCNB2*	51247755	51272733	189,885			
		*MINDY2*	51500299	51570991	37,681			
		*RNF111*	51288677	51380159	82,459			
		*ADAM10*	51598073	51739157	135,455			
		*CLNS1A*	51841987	51843484	379,369			
X	32696913	*MAGEA*	32428998	32719060	22,147	3.67 × 10^−15^	6.39 × 10^−10^	HYK, BHB-AUC, 1 vs. 3, 1 vs. 4, 2 vs. 3 and 2 vs. 4
		*IDS*	32302897	32324344	372,569			
		*CXorf40A*	32345274	32348925	347,988			
		*TMEM185A*	32922144	32957365	225,231			
X	32726008	*TMEM185A*	32922268	32964027	196,260	2.27 × 10^−13^	1.48 × 10^−08^	HYK, BHB-AUC, 1 vs. 3, 1 vs. 4, 2 vs. 3 and 2 vs. 4
		*IDS*	32302897	32324344	401,664			
		*MAGEA*	32685989	32687155	38,853			
		*HSFX3*	32590840	32592421	133,587			
		*CXorf40A*	35613768	35614939	2,888,931			
X	57467501	*BEX3*	57269171	57270812	196,689	4.35 × 10^−15^	5.68 × 10^−10^	HYK, BHB-AUC, 1 vs. 3, 1 vs. 4, 2 vs. 3 and 2 vs. 4
		*CXorf57*	56957121	57062226	405,275			
		*TCEAL9*	57244995	57246736	220,765			
		*PLP1*	57864748	57881720	397,247			
		*RAB9B*	57864581	57926698	397,080			
		*MORF4L2*	57744023	57754954	276,522			
		*GLRA4*	57785197	57798407	317,696			
X	68194066	*HTR2C*	67986710	68083180	110,886	4.42 × 10^−15^	3.30 × 10^−10^	HYK, BHB-AUC, 1 vs. 3, 1 vs. 4, 2 vs. 3 and 2 vs. 4
		*LHFPL1*	68428002	68491176	297,110			
		*SNORA35*	67792018	67792065	402,001			
		*AMOT*	68598671	68657930	404,605			
		*RTL4*	68212448	68289758	18,382			
		*RBMX2*	68131773	68142135	51,931			
X	85371223	*SLC7A3*	85010185	85015654	361,038	2.70 × 10^−12^	1.57 × 10^−07^	HYK, BHB-AUC, 2 vs. 3 and 2 vs. 4
		*DLG3*	85245062	85298606	72,617			
		*GDPD2*	85309754	85319162	52,061			
		*KIF4A*	85321378	85449490	0			
		*P2RY4*	85481339	85482436	110,116			
		*AWAT1*	85499886	85508546	128,663			
		*DGAT2L6*	85556422	85577736	185,199			
		*IGBP1*	85588914	85635721	217,691			
		*EDA*	85708003	86099973	336,780			
		*IL2RG*	84816145	84819841	551,382			
		*IKZF5*	85444282	85447321	73,059			
X	95872578	*HSD17B10*	96267144	96269467	396,889	4.03 × 10^−16^	2.10 × 10^−10^	HYK, BHB-AUC, 1 vs. 3, 1 vs. 4 and 2 vs. 4
		*MAGED4B*	95546166	95553643	318,935			
		*GPR173*	95933520	95953706	60,942			
		*KDM5C*	96041773	96072571	169,195			
		*SMC1A*	96218220	96252806	345,642			
		*RIBC1*	96253141	96266987	380,563			
		*HUWE1*	96362881	96520246	490,303			

^1^ SNP location: Exact location of the SNP within the chromosome referencing the University of Maryland (UMD) 3.1 bovine genome assembly. ^2^ Gene start and gene end: The coordinates of beginning and end for genes located on the region of influence of associated SNPs; the region of influence of each SNP was defined as 1 Mega base pair up and downstream on the UMD 3.1 bovine genome assembly. ^3^ Distance to the SNP (bp): Distance in base pairs between the SNP and the candidate gene. ^4^ Best *p*-value: Indicates the lowest *p*-value when the SNP was significant in multiple GWAS; only SNPs with a corrected FDR ≤ 0.05 were analyzed to diminish the probability of Type I error. ^5^ Shows all different explanatory variables where the SNP was significant with a corrected FDR ≤ 0.05; GWAS: Hyperketonemia (HYK/dichotomous), BHB-AUC (continuous response), Group 1 vs. Group 2 (dichotomous), Group 1 vs. Group 3 (dichotomous), Group 1 vs. Group 4 (dichotomous), Group 2 vs. Group 3 (dichotomous), Group 2 vs. Group 4 (dichotomous), and Group 3 vs. Group 4 (dichotomous).

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
