# Peer review of "Longitudinal Phenotypes Improve Genotype Association for Hyperketonemia in Dairy Cattle"

_animals, 2019, doi:10.3390/ani9121059_

Round 1

Reviewer 1 Report

This manuscript is a well written, precise and concise description of a well designed study to explore a complex metabolic disease and its genetic associations.

Although the results of the study do not provide any immediate actionable interventions at mitigating the disease, they do provide the scientific community with information on how to further explore it.

Even after careful reading, I can not provide any suggestions for improvement of the manuscript.

Reviewer 2 Report

Within their manuscript, Yepes et al. present genomic analysis of BHBH and NEFA levels in early lactation as indicators of meatbolic stability / energy status. The manuscript is well wrirren and the results are clearly presented. i only have two comments:

The authors define phenotypic classes based on combinations of high / low AUC for BHBA and NEFA. Did they also try bivariate analyses, which might give a more detailed view? The sample size might be too small, though. Could the authors please comment on that? I think, there should be some correction for milk yield , i.e. energy output. Was thatv tested?

Reviewer 3 Report

Dear Authors,

I have reviewed the manuscript entitled 'Longitudinal phenotypes improve genotype association for hyperketonemia in dairy cattle' by Leal Yepes et al.

The manuscript presents a study on hyperketonemia in dairy cattle.
The study seems well conducted, but suffers from low number of individuals involved. While it is hard to determine what should be the lower number of individuals involved, the large standard errors for heritability estimates and potentially large number of false positive discoveries in the GWAS (by looking at the Q-Q plots) suggest that there is not sufficient power to detect associations. As for the heritability estimates, the Authors don't report about the software and algorithm used, so it is hard to interpret those results. But standard errors show that confidence intervals escape the plausible heritability values (0 to 1), which is not a good sign.

I suggest the Authors to acknowledge this limitation, and do not take the heritability estimates as reliable.

In-text comments:

Lines 118-121: I don't see the need to have this paragraph here.
Line 132: please report a reference for the 'trapezoidal rule'.
Lines 201-205: Please report the software and algorithm used to estimate variance components.
Line 203: Please replace 'variance caused by the environment' with 'is the estimate of environmental variance'.

Round 2

Reviewer 3 Report

I appreciate the time that the Authors took in addressing my concerns.